# "Scoliosis 3D"—A Virtual-Reality-Based Methodology Aiming to Examine AIS Females' Body Image

Ewa Misterska [1,*], Filip Górski [2], Marek Tomaszewski [3], Pawel Buń [2], Jakub Gapsa [2], Anna Słysz [4] and Maciej Głowacki [5]

1 Department of Pedagogy and Psychology, University of Security, 60-778 Poznan, Poland
2 Institute of Materials Technology, Poznan University of Technology, 61-138 Poznan, Poland
3 Department of Spine Disorders and Pediatric Orthopedics, Poznan University of Medical Sciences, 61-545 Poznan, Poland
4 Department of Psychology and Cognitive Sciences, Adam Mickiewicz University, 60-547 Poznan, Poland
5 Department of Pediatric Orthopaedics and Traumatology, Poznan University of Medical Sciences, 61-545 Poznan, Poland
* Correspondence: ewa.misterska@wsb.net.pl; Tel.: +48-61-8-510-51; Fax: +48-61-642-15-99

**Abstract:** Modern techniques such as virtual-reality (VR) tasks might offer a unique method for eliciting state-variable fluctuations in body satisfaction and associated behaviors. The study aim was to develop the application of biometric avatars in VR as a useful tool to investigate changes within body representation in adolescent idiopathic scoliosis (AIS). All the avatars were created on the basis of 3D scans of bodies of real female patients with thoracic scoliosis, of 12–18 years of age, consecutively selected for brace treatment or posterior correction and fusion. A 3D, white-light LED scanner was used. The models were rigged using 3DS Max software, to enable the possibility of human-type interaction and animation. The "Avatar Scoliosis 3D" is an innovative 3D, interactive-XR application, loosely based on the virtual-mirror concept, and contains a number of predefined avatars, each with a different Cobb angle. It is possible to change a selected avatar to one with a different Cobb angle (lower or higher), should the patient decide the visualization of the original is incompatible with their own perception. In conclusion, the possible application of biometric avatars in VR as a useful tool to investigate changes within body image in AIS was proposed.

**Keywords:** adolescent idiopathic scoliosis; spine deformity; body image; XR (extended reality); virtual mirror; avatars

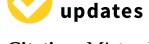



## 1. Introduction

Adolescent idiopathic scoliosis (AIS) is one of the most frequently occurring deformities of organ motion, with a prevalence of 2–3% in children and adolescents [1]. AIS is characterized by a three-dimensional deformity which occurs across three planes: frontal (lateral bend), sagittal (physiological deformation resulting in thoracic kyphosis and lumbar lordosis) and transverse (associated with the rotation and translation of vertebrae) [1]. Scoliosis progression tends to occur more frequently in girls and, therefore, girls need treatment more often than boys [1]. Scoliosis with a high Cobb angle can result in the reduction of a patient's physical capacity, neurological disorders and, in extreme cases, in cardio-respiratory failure and premature death [1].

Additionally, trunk deformity in AIS is a significant cosmetic, psychological, and social problem, particularly in cases of trunk decompensation and rib hump, especially of a high angle [1–5]. A steadily increasing angle of curvature of up to a 45–50 degree Cobb angle, neurological disorders and pain are indications for operative treatment, as are, in some cases, aesthetic reasons connected to rib hump or lumbar curve [1]. In particular, individuals with progressive idiopathic scoliosis may have several pronounced body deformities, including scapular and rib prominence, uneven shoulders, and an asymmetric waistline. According

to some authors, these deformities can significantly influence personality traits, levels of self-esteem, and self and body image which, among others, are decisive factors affecting the quality of life of adolescent patients [6–9]. Specifically, Auerbach et al. [9] indicated greater back-related body-image disturbances in patients with scoliosis compared with healthy controls.

In the view of challenges related to the need for objective measurement of body representation, some modern techniques, such as virtual-reality (VR) tasks might offer a unique method for eliciting state-variable fluctuations in body satisfaction and associated behaviors by allowing near-perfect control over environmental factors. VR has been used to study social behaviors toward, e.g., stigmatized groups; for example, researchers have measured interpersonal distance as an indicator of participants' level of stigmatization toward avatars who are perceived to be HIV-positive [10,11]. Furthermore, one study recently investigated the use of virtual embodiment to reduce self-criticism in a non-clinical sample of young women, suggesting that VR may show promise as a tool to reduce the detrimental effects of self-stigmatization among members of marginalized groups, including people with visual body deformities [12].

To the best of our knowledge, this would be the first study to create biometric avatars in VR to investigate changes within body representation in AIS. The concept of VR tasks is modeled after the design of Mölbert et al. [13], with our original modification dedicated to AIS patients. This innovative method would allow realistic manipulation of avatar body shape and the investigation of perception of other bodies in a well-controlled environment. Moreover, as this field of research develops, it would be important to consider how virtual interactions and avatar embodiment, among other factors, may influence body-shape perception within a virtual setting. Developing a nuanced understanding of the perception of the silhouette, in the context of virtual spaces, incorporating such themes as embodiment, interaction with avatars, and one's self-perspective in VR, may allow the development of universal, disseminable virtual interventions to improve body satisfaction in the future.

VR-related analyses can enable a better description and understanding of the somatic-disease mechanism influencing adolescent patients and the treatment and intervention methods applied. In conclusion, we aimed to create a project methodology, concerning the application of biometric avatars in VR, as a useful tool to investigate changes within body representation in AIS. An experimental procedure, using a library of realistic, virtual-3D avatars, to allow for realistic scoliosis-related body-deformity manipulations and a naturalistic scenario presentation of these avatars, was also developed.

## 2. Methods

Firstly, to record AIS patients' body shape, a full-body scanning system was used. Then, based on three-dimensional (3D) body scans, realistic virtual-3D bodies (avatars) that were varied through a range of ±10 of the participants' Cobb's angle, were created.

### 2.1. Procedures

This study group was comprised of patients with AIS consecutively selected for Cheneau-brace treatment or posterior correction and fusion. All were recruited from one academic center, the Department of Pediatric Orthopedics. The additional inclusion criteria were: female, 12–18 years of age, thoracic scoliosis. Criteria set by The Scoliosis Research Society on the location of apex were followed [14].

Only a homogenous group of females with thoracic scoliosis was taken into account, to limit the impact of other variables that might have influenced the study results. In particular, it was revealed in a study examining AIS patients following surgical treatment, that, amongst other radiological data, only the value of the thoracic-apical translation had a significant influence on the perception of trunk deformity. Furthermore, it was found that a higher thoracic-apical translation decreases the probability of positive perception of body shape following surgical treatment [15].

Girls in whom other diseases leading to trunk deformity or serious medical conditions were diagnosed, were excluded.

At the beginning of this phase, each AIS female who fulfilled the inclusion criteria, and her legal guardian, were fully informed of the aim of the research, and that, based on their body scans, realistic virtual-3D bodies (avatars) would be created. Then, they were assured of anonymity, following which they gave their informed consent. An orthopedic surgeon, trained in body-scanning system support, was available should participants or their legal guardians require explanation or clarification.

To standardize the body-scanning process, all selected patients were dressed in a unified breast band. Then, three body scans in the T-pose, A-pose and the neutral pose were taken, resulting in three high-polygon meshes and three RGB images for texture generation.

During this study period, 32 patients who fulfilled the inclusion criteria, were recruited. Then, a team comprised of experienced orthopedic surgeons, including surgeons trained in the body-scanning system, reviewed the available body scans of participating patients in order to qualify the images for inclusion in a library of realistic virtual-3D avatars.

It was assumed, that the avatars should be presented giving several ranges of $\pm 10$ Cobb angle. Thus, patients with the following ranges of Cobb angle were selected: (1) Cobb angle of 10–19; (2) Cobb angle of 20-; (3) Cobb angle of 30–39; (4) Cobb angle of 40-; (5) Cobb angle of 50–59; (6) Cobb angle of 60–69; (7) Cobb angle of 70–79.

### 2.2. Body-Scanning System

All the avatars were created on the basis of the aforementioned 3D scans of bodies of real patients. A structured-light 3D scanner was used for that purpose. A white-light LED scanner was used, with a 5 MPix sensor and measurement area of 2200 × 1500 × 500 mm. Measurement resolution was 0.85 mm (1 pt/mm$^2$) and measurement uncertainty declared by the producer was 1 mm. The 3D scanner was produced by SMARTTECH (Łomianki, Poland).

### 2.3. Graphic Processing of 3D Scans

The results of the 3D-scanning process provided 3D static triangular meshes of patients' bodies. The models were rigged using 3DS Max software 2022, to enable possibilities of human-type interactions and animations. The avatar models, in FBX format, were imported into Unity software and appropriately converted, adding materials and textures, and skeletal animation was also configured (skinned-mesh renderers were used, with an appropriate bone rig system).

## 3. Results

Based on a library of realistic virtual-3D avatars, the "Avatar Scoliosis 3D" application was developed, to allow for realistic scoliosis-related body-deformity manipulations and a naturalistic-scenario presentation of these avatars.

### 3.1. Technical Setup

The "Avatar Scoliosis 3D" is an innovative 3D, interactive-XR application, loosely based on the virtual-mirror concept (Figure 1). Virtual mirrors are augmented-reality (AR) applications, in which the user sees their own image (either a digital still or via a video-camera), extended (augmented) with certain 3D interactive visualizations. The most common application for the virtual mirror is in sales [16], but there are known medical uses—in therapy, for the treatment of phantom pain after limb amputation [17] or in the design of personalized orthopedic devices [18,19]. The authors therefore assume that this concept will prove feasible for the proposed therapeutic procedure.

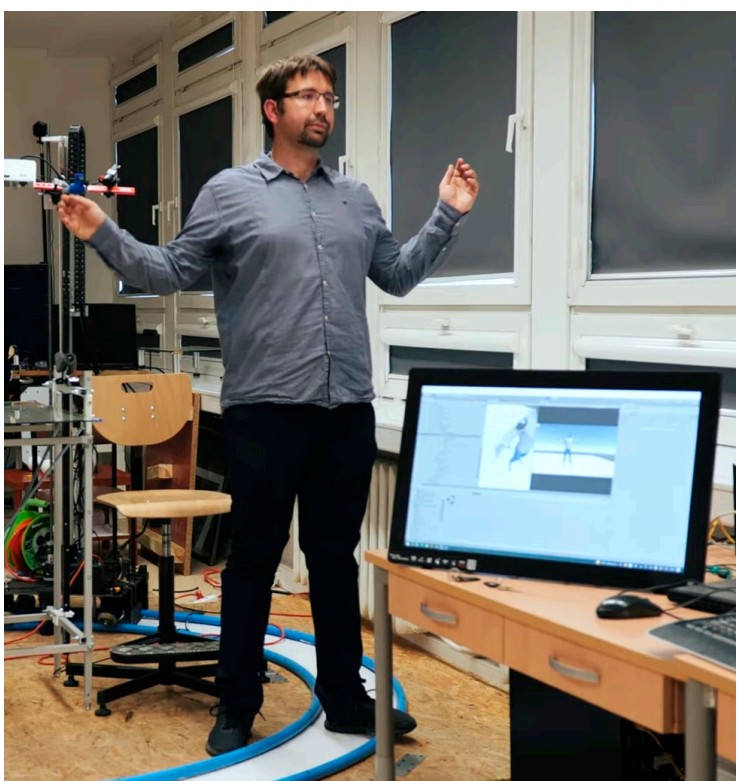

**Figure 1.** Concept of virtual mirror with body avatar.

The application itself is based on a full-body VR tracking system provided by the HTC company. The low-cost Vive Tracker devices (Figure 2) are employed to perform real-time measurements of a patient's body. At the same time, a virtual, digital representation of the patient's body (an avatar) re-creates the movement, enabling the patient to look at their own body and posture, both in a static view and in movement. Camera angles and zoom can be slightly altered by the investigator (not by the patient). As such, patients can see and perceive themselves moving. The setup can be classified both as a VR system (although without a headset), and as an AR system.

The application is constructed using Unity engine, version 2019.4.37 (Unity Technologies, San Francisco, CA, USA). VR interactions and camera movement are available through the SteamVR plugin for Unity engine, version 2.7.3 (Valve Corporation, Bellevue, WA, USA), available through the Unity Asset Store platform, working with the SteamVR desktop application (available through the Valve Steam platform, version 1.24). Vive Trackers are incorporated through the Vive Input Utility plugin for Unity Engine, version 1.17.0 (HTC Corporation, New Taipei City, Taiwan), available likewise through the Unity Asset Store platform. In addition, the Final IK Unity plugin, version 2.2 (RootMotion, Tartu, Estonia) is used to simulate avatar movement, applying input from the trackers. All the software products used do not have specified hardware requirements, other than having appropriate hardware up and running in the same system.

The application itself is a Windows PC application, running on a computer equipped with Vive Trackers and Vive Base Stations. Thanks to a special configuration of the SteamVR software with a so-called "null HMD", no VR headset is required to run the system (it would usually be required, to gather data from the trackers). The use of such a system for body tracking, instead of a professional motion-capture system, is viable in terms of the available literature studies [20] and it will allow, in the future, for easier scalability of the system. The Vive-Tracker setup is also significantly cheaper than the professional mocap systems. The application interface consists of two main screens—the initial screen is used to input basic patient data, and to allow them to select the appropriate avatar. In the second

screen, the selected avatar is displayed in a neutral environment, and the patient is able to perform movements using the body-tracking system.

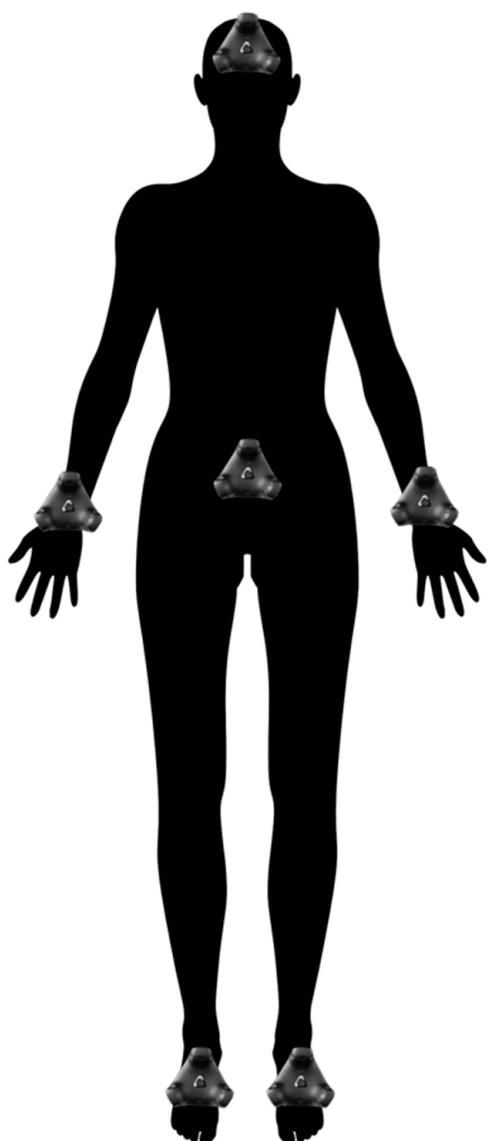

**Figure 2.** Vive Trackers—proposed setup on patient's body.

The "Avatar Scoliosis 3D" application contains a number of predefined avatars, with different Cobb angles. It is possible to change a selected avatar to one with a different Cobb angle (lower or higher), should the patient decide the visualization of the original is incompatible with their own perception. Body tracking is turned on, and the patient is able to see the selected avatar in motion. The patient then selects the avatar which is the closest to their actual body shape, and then chooses the desired perception of their body from those available.

### 3.2. Stimulus-Image Generation

The avatar is always displayed in a neutral environment—a room with brightly-colored walls and a wooden floor, to avoid distraction. The patient is equipped with a total of six trackers:

One for each hand (two in total)
One for each foot (two in total)
One for the head

One for the waist.

The visual angle of the whole avatar is at first 0°. However, participants are told that they are able to look around freely, if they need to.

In Experiment 1 and 2, participants stand at a 2 m (±25 cm) distance from the screen, as though facing themselves in a mirror. The scene is presented as a flat, large-screen immersive display, onto which the stimuli are projected using an Acer H6512BD 3D projector.

### 3.3. Experimental Procedures

Based on the "Avatar Scoliosis 3D" application, an experimental procedure using a library of realistic virtual-3D avatars to allow for realistic scoliosis-related body-deformity manipulations and a naturalistic-scenario presentation of these avatars was developed.

In general, the investigator provides all participants with a verbal description of the purpose of the VR tasks, via the "Avatar scoliosis 3D", explaining the components of the VR experience. To sum up, participants are told that they will be participating in a study examining their perception of actual and desired body shape.

In particular, the procedure comprises two experimental sessions (E1 and E2), in which avatars are presented on an immersive life-size stereoscopic display that mimics in virtual reality the situation of looking at oneself in a mirror. Therefore, participants will complete two method-of-adjustment tasks (MoA), the first referring to current body shape (E1) and the second referring to ideal body shape (E2).

At the beginning of the procedure, each participant is informed that, based on real-patient body scans, a set of bodies will be generated. Participants are also told that the presented avatars can either represent their body exactly or be a modified version of their body, gradually made less or more deformed.

In the E1 and E2 tasks participants are shown each avatar, corresponding to the following ranges of Cobb's angle that were selected before: (1) Cobb angle of 10–19; (2) Cobb angle of 20–29; (3) Cobb angle of 30–39; (4) Cobb angle of 40–49; (5) Cobb angle of 50–59; (6) Cobb angle of 60–69; (7) Cobb angle of 70–79 (Figures 3 and 4). Participants are shown the avatar with no time limit, and have to adjust it to their current [E1] or ideal [E2] body shape [15]. Then, body tracking is turned on and the patient is able to see the selected avatar in motion.

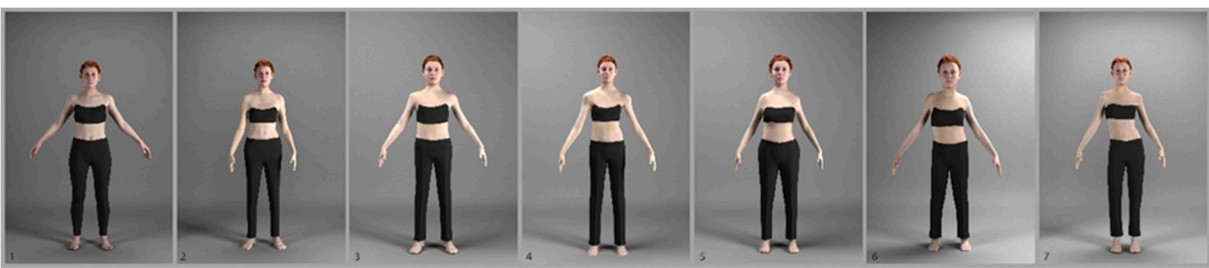

**Figure 3.** 3D scoliosis avatars (front view).

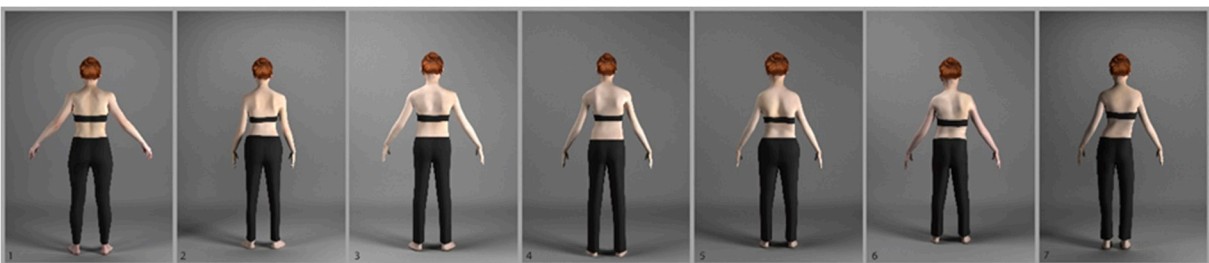

**Figure 4.** 3D scoliosis avatars (back view).

Avatar no. 1 presents a Cobb angle of 10–19; no. 2 presents a Cobb angle of 20–29; no. 3 presents a Cobb angle of 30–39; no. 4 presents a Cobb angle of 40–49; no. 5 presents a Cobb angle of 50–59; no. 6 presents a Cobb angle of 60–69; and no. 7 presents a Cobb angle of 70–79.

At the beginning of each experiment, the investigator provides the instruction, after which the avatar appears. In E1, the instruction given is as follows: "Please adjust the body until it matches your current body!". In E2, the instruction is modified to "Please adjust the body shape on screen until it matches your ideal, desired body shape". During the session, it is possible to change the selected avatar to one with a different Cobb angle (lower or higher) if the patient's perception is incompatible with the visualization of the initial avatar.

Although there is no set time limit to a session, participants are instructed to rely on their instinct and to not linger too long over a decision.

Furthermore, after the session, participants complete a questionnaire in which they are asked to rate on a Likert scale from 1 (not at all) to 7 (very much) a) how similar they perceived the two finally selected avatars to be (overall impression) and b) whether, as the avatars were presented on an immersive life-size stereoscopic display mimicking looking at a mirror reflection of themselves in virtual reality, they could identify with the presented avatars. Piryankova et al. [21] observed such ratings to be sensitive to dissimilarities between avatar and participant.

The experimental procedure for control, healthy female adolescents was also developed, for comparative purposes. The procedure comprises an experimental session with Experiment 1 (E1). Specifically, participants complete one method-of-adjustment task (MoA), referring to the estimation of the current (E1) body shape. The only instruction is, "Please adjust the body until it matches your current body!". After the session, participants complete a questionnaire, which asks them to rate on a Likert scale from 1 (not at all) to 7 (very much) a) whether, as the avatars were presented on an immersive life-size stereoscopic display mimicking looking at a mirror reflection of themselves in virtual reality, they could identify with the presented avatars.

*3.4. VR-Task Analysis*

From different experimental tasks conducted with AIS patients both pre-and post- surgical treatment, the following indicators are extracted: (1) the degree of inaccuracy/distortion of the estimated body shape at the time of E1, as compared to participants' actual body shape; (2) the degree of inaccuracy/distortion of the desired body shape at the time of E2, as compared to participants' actual body shape; (3) the discrepancy between the estimated body shape at the time of E1, as compared to participants' desired body shape at the time of E2; (4) a comparison of the degrees of the distortion described in points 1, 2 and 3 above obtained in AIS patients before and following surgical treatment; (5) a comparison of the degrees of distortion as described by points 1, 2 and 3 above obtained in the scoliosis sample following any therapeutic intervention and scoliosis control sample.

The following indicators are extracted in the control sample: (1) the degree of inaccuracy/distortion of the estimated body shape at the time of E1, as compared to participants' actual body shape; (2) a comparison of the degrees of the distortion as described in point no. 1 above, in the control sample and in the scoliosis sample; (3) a comparison of the degrees of the distortions as described in point no. 1 above in the control sample and in the scoliosis sample, following any therapeutic intervention; (4) a comparison of the degrees of the distortion as described in point no. 1 above in the control and the scoliosis control samples.

## 4. Discussion

As far as the authors are aware at this time, the proposed methodological project would be the first study concerning the application of biometric avatars in VR to investigate changes within body representation in AIS. It was developed in response to the fact that scoliosis patients following surgical treatment might further overestimate body deformities, e.g., the magnitude of rib hump, scapular and rib prominence, uneven shoulders or

asymmetric waistline, and at the same time, experience bod- image disturbances in the long term. In addition, the project was created in the view of challenges related to the need for objective measurement of body representation.

The results of the application of AIS avatars may have important implications for the development of standards of body-image-disturbance (BID) treatments in AIS. Promising evidence suggests that implicit interventions aimed at correcting maladaptive automatic processes may be effective at improving body image in other clinical samples [22,23].

Spina- fusion surgery with instrumentation often successfully reduces severe curves and minimizes the risk of curve progression [1]. However, even though surgery is technically successful and despite the physical benefits of spinal-fusion surgery, which include the preservation of pulmonary function and the prevention of osteoarthritis [24,25], AIS patients' satisfaction with surgical outcomes can be lower than may be expected, as it is often based on the perception of postoperative cosmesis of the back and shoulders [26].

Considering the issue of body representation in AIS, it has long since been conceptualized as a hierarchical construct with different components [27]. As there is no clear evidence for any such distinction, a dimensional model has recently been developed [28,29]. In this model, body representation is a conglomerate of multiple body representations that can be characterized in terms of how explicit v. implicit they are, and the extent to which they are perceptual v. conceptual. The body representations are informed by different senses and modalities, such as vision, proprioception or even social comparison, and can be integrated into higher-level representations [13].

Not much is known about changes in AIS patients' body representation over a longer time frame. Noonan et al. have shown a more negative body image that may persist for several years in surgical AIS patients [30]. In addition, they found that the perception of self-image might also deteriorate with time. Interestingly, Koch et al. indicated that patients with scoliosis who reported neutrality with the postoperative cosmesis, experienced multiple coping problems and a more critical, negative view of themselves. They also continued to perceive themselves as less attractive than their peers [31].

Several studies concerning BID in clinical samples, e.g., patients treated due to anorexia nervosa (AN), observed that those patients overestimate their body size in different visual-size-estimation tasks [32–34] as well as in non-visual measures [35]. Researchers also indicated that women with BID neither see their own body nor other weight-matched persons differently than controls, but they evaluate them differently in terms of what weight is desirable [13]. In the current project it is assumed that a similar phenomenon, regarding body-shape estimation, might occur within the AIS population. It should be underlined that results concerning the issue of body-shape evaluation seem contradictory. On the one hand, it was revealed that body dissatisfaction is related to the altered (disfigured) shape and the weight of the individual, but, on the other hand, improvement in body shape and weight loss does not necessarily modify (decrease) body dissatisfaction [36].

This contradiction might be explained by the allocentric-lock hypothesis (ALH) [37]. It suggests that maintenance of body dissatisfaction may be caused by a primary disturbance in the way the body is experienced and remembered: patients may be locked into an allocentric (observer view), disembodied negative memory of the body that is not updated by contrasting egocentric representations driven by perception [38]. In other words, AIS patients may be locked onto a negative image of their bodies, a mental representation that perception is unable to update even after significant body-shape improvement following surgical treatment.

To sum up, it is still unclear under which circumstances patients with AIS overestimate the perception of scoliosis-related body deformation, experience body-image disturbances further after operative treatment, and, finally, how this overestimation is characterized. In the clinical context, these findings suggest that patients with AIS might need support in changing the perception of their desired body shape and in feeling positive about the cosmetic results of scoliosis surgical treatment.

*Future Research Implications*

The next stage of this project, is a two-fold longitudinal assessment of changes in the psychosocial functioning of female patients with AIS before and after completion of surgical treatment and implementation of cognitive-behavioral-therapy (CBT) interventions. A control group of healthy females will also be selected for comparative purposes. In particular, the following factors will be assessed: body-image disturbances, body-shape estimation, and mental health according to the following criteria: emotional symptoms, behavioral disorders, hyperactivity, concentration disorders, problems with peer relations and pro-social behavior, and objective aesthetic evaluation of trunk deformity performed by doctors.

Considering other research implications, the VR set could be used in exposure-based treatments. Specifically, AIS patients with high BID could practice behavioral skills in VR and then apply them to real environments. The results of the proposed research project would also indicate that immersive virtual environments may offer unique advantages to measure body-disfigurement stigma at the behavioral level. Such methods may inform the development of more robust interventions to reduce stigmatized beliefs about persons with easily recognizable body deformities, which prove difficult to modify through more traditional, dissonance-based approaches [39,40].

## 5. Conclusions

In conclusion, the possible application of biometric avatars in VR as a useful tool to investigate changes within body image in AIS was proposed. The reported experimental procedure, utilizing a library of realistic virtual-3D avatars, can allow for realistic scoliosis-related body-deformity manipulations and a naturalistic-scenario presentation of these avatars.

To sum up, this methodological project is expected to provide new knowledge that, in the future, should inspire further applied research, and could make a significant contribution to the development of guidelines for good interdisciplinary rehabilitation of AIS patients undergoing spinal fusion.

**Author Contributions:** Conceptualization, E.M., F.G., P.B. and M.G.; methodology, E.M., F.G., P.B., and M.G.; software, F.G. and J.G.; validation, E.M., F.G. and M.G.; formal analysis, E.M., F.G. and M.G.; investigation, M.T.; resources, E.M., F.G. and M.G.; data curation, M.T. and M.G.; writing—original draft preparation, E.M. and F.G.; writing—review and editing, A.S. and M.G.; visualization, F.G. and J.G.; supervision, M.G.; project administration, E.M. and M.G.; funding acquisition, E.M. and M.G. All authors have read and agreed to the published version of the manuscript.

**Funding:** This project was funded by the National Science Centre, Poland (grant number: 2017/27/B/NZ5/02109).

**Institutional Review Board Statement:** The project was conducted according to the guidelines of the Declaration of Helsinki, and approved by the Bioethical Commission at Poznan University of Medical Sciences (No. 695/18, No. 800/22) and by the Center for Safety Research at the University of Security in Poznan (No. 001/2018).

**Informed Consent Statement:** Informed consent was obtained from all subjects involved in the study.

**Data Availability Statement:** The data presented in this study are available on request from the corresponding author. The data are not publicly available, due to privacy.

**Acknowledgments:** This work was supported only by the National Science Centre, Poland. We would like to warmly thank Przemyslaw Matejko for graphic processing of 3D scans.

**Conflicts of Interest:** The authors declare no conflict of interest.

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
