# Peer review of "“Scoliosis 3D”—A Virtual-Reality-Based Methodology Aiming to Examine AIS Females’ Body Image"

_applsci, doi:10.3390/app13042374_

Round 1

Reviewer 1 Report

Dear authors,

The manuscript is very interesting but English editing needs to be addressed. There are many typing oversights, I recommend the revision of the manuscript by a native person or by an English Editing Service. 

In line 290 BID stands for body image disturbnces while in line 317 body image disorders. Please revise also all the abbreviations.

The discussion should be revised while this paragraph is not clear at all “Spinal fusion surgery with instrumentation often successfully reduces severe curves 293 and minimizes the risk of curve progression [4,5]. However, the cosmesis of the back and 294 shoulders, is often the most critically important factor to adolescents with idiopathic 295 scoliosis, making the patient’s perception of postoperative silhouette an indicator of sat- 296 isfaction with the surgical result [26]. For that reason, unfortunately, surgeons’ technical 297 successes do not necessarily translate into AIS patient satisfaction with surgical outcomes. 298 To sum up, patient satisfaction with the surgical result appears unrelated to the physical 299 benefits of spinal fusion surgery, which include the preservation of pulmonary function 300 and the prevention of osteoarthritis [27,28].

Author Response

Response to 1st reviewer can be seen in the attachement /as Word file.

Reviewer 2 Report

Thank you for your proposal and research!

I appreciate the precision and detailed presentation of the work.

Just one note:

lines 122 and 266 must be italic not bold and with the same font...

In my opinion the paper is ok for publication.

Author Response

Response to 2nd Reviewer can be seen in the attachement.

Reviewer 3 Report

Novel study.

Author Response

Response to 3rd Reviewer can be seen in the Word file.

Reviewer 4 Report

Review for Applied Sciences – “Scoliosis 3D” – a virtual reality-based methodology aiming to 2 examine AIS females’ body image

Recommendation: Major revision

This conceptual paper explores the potential of using XR to evaluate the body image of AIS patients. It offers a detailed method for creating a biological avatar using virtual reality, as well as an interactive use of the VR avatar as an XR, which could be used in a future clinical assessment of AIS patients.

Although the concept authors suggested was generally clear and original, several concerns should be addressed.

1.         Introduction line 31-35: As introduction, the importance of scoliosis treatment was enough illustrated and these lines did not affect the content of this manuscript. Especially line 34-36 about osteopenia and AIS was not seemed to be matched with reference 2 and 3. Please kindly double-check the content of papers and descriptions here.

2.         Method line 90-97: It is reasonable to include patients with thoracic AIS because the apical translation is a major factor influencing their appearance questionnaire. However, I think the sentence structure in this section makes it difficult for readers to understand. Please paraphrase them to make the more concise.

3.         Result line 160-164: What would you like to convey with the details of “(Windows PC)”, “(a Unity asset)”, and “Final IK plugin”? Are there any more details such as version or memory requirement?

4.         Result line 204-212: I could not understand these procedures and tasks. Kindly paraphrase these sentences for better understanding.

5.         Results 223: Table 1 is missing; is it a typo in Fig3? Also, Figs 3 and 4 do not seem to be ordered by severity of scoliosis. For example, Fig 3: 7 is clearly milder than Fig 3: 6.

6.         Fig 4: the number 7 is missing in the figure.

7.         Result line 231-244: These lines were difficult to follow due to multiple unstructured paragraphs. Please kindly summarize the E1 and E2 tasks more clearly and concisely.

8.         Line 371-375: Figure legends does not match to the original one in the manuscript.

Author Response

Response to 4th Reviewer can be seen in the Word file.

Round 2

Reviewer 4 Report

Dear authors,

I have carefully reviewed the revised manuscript entitled "Scoliosis 3D” – a virtual reality-based methodology aiming to examine AIS females’ body image". I would like to commend the authors for your efforts in addressing the concerns that I had raised in my previous review.

While most of my comments have been addressed satisfactorily, there is one concern that I believe still needs attention. Specifically, in the revised discussion section at lines 355-362, I have noticed that two of the citations (24 and 25) included do not appear to be directly relevant to the topic of AIS and osteopenia. 

I would be grateful if the authors could kindly provide additional explanation for these citations in this section.

Thank you for the opportunity to review this manuscript.

Author Response

Authors' response to the 2nd review

Dear Editor,

We would like to thank you for giving us the opportunity to revise our manuscript. We have carefully read all the remarks made by the referee and addressed all of the reviewer’s comments.   

In this cover letter we have given specific details on how we addressed each comment. All the passages that we have changed or added to the manuscript, are marked up using the “Track Changes” function".

Once again, thank you for reconsidering our manuscript.

REVIEWER’S COMMENTS
Reviewer #4:

Comments and Suggestions for Authors

„Dear authors,

I have carefully reviewed the revised manuscript entitled "Scoliosis 3D” – a virtual reality-based methodology aiming to examine AIS females’ body image". I would like to commend the authors for your efforts in addressing the concerns that I had raised in my previous review.

While most of my comments have been addressed satisfactorily, there is one concern that I believe still needs attention. Specifically, in the revised discussion section at lines 355-362, I have noticed that two of the citations (24 and 25) included do not appear to be directly relevant to the topic of AIS and osteopenia. 

I would be grateful if the authors could kindly provide additional explanation for these citations in this section.

Thank you for the opportunity to review this manuscript.”

We would like to warmly thank the Reviewer for his careful analysis of our paper. We concur with the Reviewer’s opinion, that, in fact, two of the citations (24 and 25) included do not appear to be directly relevant to the topic of AIS and osteopenia. Therefore, finally, we have decided to delete them.